# Topological Spin Liquid with Symmetry-Protected Edge States

Yan-Cheng Wang,[1] Chen Fang,[1] Meng Cheng,[2] Yang Qi,[3, *] and Zi Yang Meng[1, †]

[1]*Beijing National Laboratory for Condensed Matter Physics,*
*and Institute of Physics, Chinese Academy of Sciences, Beijing 100190, China*
[2]*Department of Physics, Yale University, New Haven, CT 06511-8499, USA*
[3]*Department of Physics, Massachusetts Institute of Technology, Cambridge, MA 02139, USA*
(Dated: August 28, 2017)

Topological spin liquids are robust quantum states of matter with long-range entanglement and possess many exotic properties such as the fractional statistics of the elementary excitations. Yet these states, short of local parameters like all topological states, are elusive for conventional experimental probes. In this work, we combine theoretical analysis and quantum Monte Carlo numerics on a frustrated spin model which hosts a $\mathbb{Z}_2$ topological spin liquid ground state, and demonstrate that the presence of symmetry-protected gapless edge modes is a characteristic feature of the state, originating from the nontrivial symmetry fractionalization of the elementary excitations. Experimental observation of these modes on the edge would directly indicate the existence of the topological spin liquids in the bulk, analogous to the fact that the observation of Dirac edge states confirmed the existence of topological insulators.

## I. INTRODUCTION

Topological spin liquids [1–3] are new states of quantum matter beyond the Landau-Ginzburg-Wilson symmetry breaking paradigm, as they cannot be described by any local order parameters quantifying the breaking of symmetries. On the other hand, they have "intrinsic topological orders" [4–6], which are generally characterized by either long range entanglement [7, 8], by elementary excitations called anyons featuring fractional statistics [9], or by ground state degeneracies on high-genus surfaces [10]; however, none of these properties is directly accessible to experimental probes. While intrinsic topological orders do not require symmetries for protection, when they are present, anyons may carry fractional quantum numbers of the symmetries [6, 11–19]. Specially, some topological spin liquids are believed to host spin-$\frac{1}{2}$ excitations called "spinons", in sharp contrast to normal integer spin excitations like magnons. These fractional quantum numbers provide direct evidence for the existence of fractional anyons in a topologically ordered state and are therefore vital to the diagnosis of the topological order both numerically and experimentally.

Aside from the defining bulk characteristics above, a universal feature of topological states is the bulk-boundary correspondence, *i.e.*, the nontrivial topology is reflected in anomalous properties of the boundary. A well-appreciated example of this correspondence is between the symmetry-protected gapless edge modes and the bulk $Z_2$-index of topological insulators [20–22]. Unfortunately, a generic nonchiral spin liquid state, such as a $\mathbb{Z}_2$ spin liquid, does not have protected gapless edge modes [23–25]. It has been argued [26, 27], however, that if the anyons carry fractional quantum numbers of certain symmetries (lattice symmetries for example), then the symmetries can forbid condensing (gapping) the anyons on the edge, giving rise to symmetry protected gapless excitations. These edge modes are analogous to the helical edge modes in topological insulators, with the difference that here anyonic excitations, not electrons, propagate in these modes.

In this paper, we establish, via analysis and numerics, a connection between the fractional quantum numbers and the existence of symmetry-protected gapless edge modes in a $\mathbb{Z}_2$ "toric code" spin liquid, realized on a Kagome lattice. In this spin liquid, the bosonic spinon and the vison both carry nontrivial fractional symmetry quantum numbers. Through an analysis of the Luttinger liquid theory on the edge, we show that the edge must be gapless as long as time-reversal and mirror reflection symmetries are preserved. This is confirmed by state-of-art quantum Monte Carlo (QMC) simulations, where we demonstrate that there is a finite parameter region where all correlation functions on the edge decay algebraically, and that this region disappears as soon as either time-reversal or mirror reflection symmetry is broken. Our results show that the existence of protected, gapless edge modes is the hallmark of some topological spin liquids, and can hence be used to identify them in experiments.

## II. MODEL

To begin with, we introduce the model Hamiltonian, which is adapted from the Balents-Fisher-Girvin model [28]:

$$H = -\frac{1}{2} \sum_{\langle ij \rangle} \left[ (J + J') S_i^x S_j^x + (J - J') S_i^y S_j^y \right] + \frac{J_z}{2} \sum_{\hexagon} \left( \sum_{i \in \hexagon} S_i^z \right)^2 - \sum_i h_i S_i^z, \tag{1}$$

* yangqi137@icloud.com
† zymeng@iphy.ac.cn

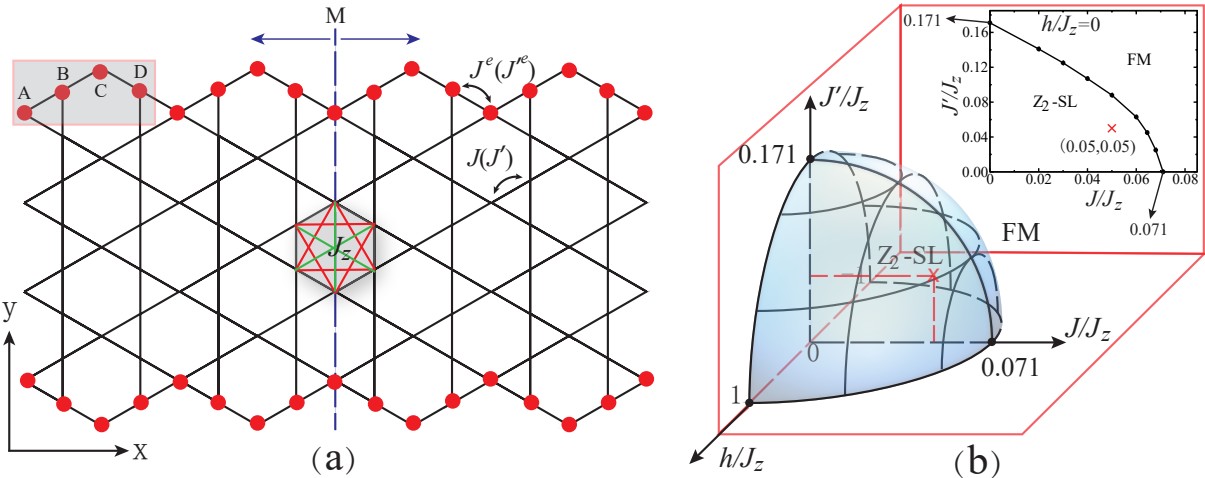

FIG. 1. (a) Kagome lattice with open boundary condition along horizontal direction. One unit cell along the edge contains four site, labelled as A, B, C and D. The interactions inside the bulk are $J$, $J'$ and $J_z$, the interactions along the edge are $J^e$ and $J'^e$. Mirror symmetry axis $M$ is denoted by the vertical dashed line. (b) Bulk phase diagram of the model in Eq.(1). $\mathbb{Z}_2$ spin liquid ($Z_2$-SL) is inside the blue volume, spanned along the three axes of $h/J_z$, $J/J_z$ and $J'/J_z$. Outside the blue volume, the system is in a ferromagnetic phase (FM), with $U(1)$ symmetry breaking on the $h/J_z$-$J/J_z$ plane and $\mathbb{Z}_2$ symmetry breaking with finite $J'/J_z$ (see text). The projection in the background is the $J/J_z$-$J'/J_z$ phase diagram at $h = 0$, and the red star is the position where the bulk parameters are chosen in this paper.

as illustrated in Fig. 1(a). Here $\mathbf{S}_i$ is a spin-$\frac{1}{2}$ operator, $\langle ij \rangle$ denotes nearest-neighbor bonds, $\varhexagon$ denotes the smallest hexagon of the kagome lattice. The original model is recovered at $J' = 0$, and becomes tractable in the limit of $J \ll J_z$, where the ground state is a gapped $\mathbb{Z}_2$ spin liquid [28]. Earlier QMC simulations confirm that the spin liquid phase is robust along the $J/J_z$ axis in Fig. 1(b), and ends at $J/J_z = 0.07076$, where the model goes through a continuous phase transition to a ferromagnetic (FM) phase [29–31].

The $J'$-term in Eq.(1) is introduced to explicitly break the continuous spin rotation symmetry along the $z$-axis down to a twofold spin rotation symmetry generated by $R_\pi = \prod_j \exp(i\pi S_j^z)$. This modification faciliates the identification of the minimal set of symmetries for the protection of the edge modes; all main results remain valid as one restores the larger U(1)-symmetry, and comments will be made whenever there is any quantitative difference. The full symmetry of the modified BFG model Eq. (1) includes $\mathbb{Z}_2$ spin rotation, time-reversal $T = \prod_j e^{i\pi S_j^y} \mathcal{K}$ with $\mathcal{K}$ being the complex conjugation, and the space group of kagome lattice. Particularly important in this work is a $\mathbb{Z}_2$ subgroup generated by a mirror reflection $M$ as shown in Fig. 1, since one can not preserve the whole point group in the presence of a physical boundary. We will also ignore translations since they play no role in the edge physics.

As shown in Fig. 1(b), our simulation unveils that with a positive $J'$ term, the $\mathbb{Z}_2$ spin liquid phase is robust and extends to the shaded region in the three-dimensional phase space (details on the QMC simulation and determination of the three-dimensional phase diagram are present in the Appendix B). Since all the points within

this region belong to the same phase, the spin liquids have the same symmetry fractionalization pattern, which we shall discuss below.

The $\mathbb{Z}_2$ spin liquid state realized in this model has three types of nontrivial anyon excitations: the bosonic spinon $e$, the vison $m$, and the fermionic spinon $\epsilon = e \times m$ (i.e. a bound state of one spinon and one vison). Both $e$ and $m$ anyons carry fractional quantum numbers of the symmetries, including spin rotation, time-reversal and lattice symmetries. Symmetry fractionalization is considered an important feature of topologically ordered systems. For example, in a $\mathbb{Z}_2$ spin liquid that has SO(3) spin rotation symmetry, a physical excitation carries integer spin quantum number, e.g. a magnon has spin 1. In the spin liquid phase, a magnon can break into two coherent spinons, each carrying a spin 1/2: this is the fractionalization of the spin quantum number, and is analogous to the charge fractionalization of electric charge in fractional quantum Hall effect [11]. For another example, if the system also has time-reversal symmetry, a magnon state under $T^2$ returns to itself ($T^2 = 1$), but when it breaks into two spinons, an individual spinon may carry $T^2 = -1$: this is the fractionalization of the time-reversal symmetry quantum number. The fractionalization of spatial symmetry such as mirror symmetry or translation symmetry is similar, but requires subtler definitions (see e.g. Ref. [14]). We summarize the symmetry fractionalization pattern of the $\mathbb{Z}_2$ spin liquid phase of the BFG model in Table I (see Ref. 27 for a derivation). In particular, it is worth noticing that $MT$ squares to minus identity on both $e$ and $m$.

TABLE I. Symmetry fractionalization in the $\mathbb{Z}_2$ spin liquid realized in the model of Eq. (1).

| Anyon | $R_\pi^2$ | $T^2$ | $M^2$ | $TM = \pm MT$ | $(MT)^2$ |
|---|---|---|---|---|---|
| $e$ | $-1$ | $-1$ | $+1$ | $+1$ | $-1$ |
| $m$ | $+1$ | $+1$ | $+1$ | $-1$ | $-1$ |

## III. EDGE STATES

Now we turn to study how symmetry fractionalization in the bulk affects the edge. Generically, the edge of a $\mathbb{Z}_2$ spin liquid can be gapped by condensing either the visons or the spinons. In the presence of symmetries, only bosons with trivial quantum numbers are allowed to condense, otherwise the symmetry defining the quantum number would be broken. If one anyon excitation carries integer quantum numbers, i.e., those allowed for magnon excitations, then one can make the anyon excitation completely "neutral" (having trivial quantum numbers) by fusing it with local spin flips, and condense this bound state. Therefore, we see that anyons carrying fractional quantum numbers is a necessary condition for obtaining a gapless edge. In our case, both spions and visons carry $(MT)^2 = -1$, so condensing either of them on the edge breaks the composite symmetry $MT$. We conclude that the symmetry fractionalization data in Table I implies that a gapped edge cannot be symmetric, but must break $MT$. Notice that the condensation of $e$ also breaks $R_\pi$ symmetry since $R_\pi^2 = -1$ on $e$. In other words, a gapless edge is then *protected* by these symmetries, in the same way the edge modes in a topological insulators are protected by time-reversal and charge conservation. We emphasize that the argument is completely general and independent of the details of the Hamiltonian near the edge.

This general but intuitive argument is supported by a concrete analysis on a low-energy effective theory for the edge. Motivated by the bulk topological field theory, we describe the edge of a $\mathbb{Z}_2$ spin liquid by the following Luttinger-liquid theory [32],

$$\mathcal{L} = \frac{i}{\pi}\partial_x\phi\partial_\tau\theta - \frac{u}{2\pi}\left[K(\partial_x\theta)^2 + \frac{1}{K}(\partial_x\phi)^2\right], \quad (2)$$

where both $\phi$ and $\theta$ are compact U(1) bosons and $u$ is the velocity on the edge and $K > 0$ is the Luttinger parameter, which controls the scaling dimensions of local operators. $e^{i\phi}$ and $e^{i\theta}$ are (non-local) operators that create vortices corresponding to spinons and visons, respectively. Symmetry fractionalization determines how $\phi$ and $\theta$ transform under symmetries, and we leave the details to the Appendix C. Note that the values of $u$ and $K$ are non-universal and depend on details of the Hamiltonian on the edge. The bulk topological order only fixes the first term in Eq. (2).

Eq. (2) describes a free gapless boson. There can be additional vortex terms as shown below, which determine whether the Luttinger-liquid theory is gapped or not:

$$\mathcal{L}_v = \sum_{p=1}^\infty \lambda_{e,p}\cos(2p\phi + \alpha_{e,p}) + \sum_{q=1}^\infty \lambda_{m,q}\cos(2q\theta + \alpha_{m,q}). \quad (3)$$

Here, $\lambda_{e,p}$ and $\lambda_{m,q}$ terms create $2p$ spinons and $2q$ visons, respectively. The operators $e^{2ip\phi}$ and $e^{2iq\theta}$ have scaling dimensions $p^2K$ and $q^2/K$, respectively. Therefore, the term $\lambda_{e,p}$ becomes relevant when $K < 2/p^2$, and the term $\lambda_{m,q}$ becomes relevant when $K > q^2/2$. Notice that because of the nontrivial commutation relation between $\phi$ and $\theta$, as soon as either of the fields condenses, the other becomes confined. Therefore, the edge is gapless if and only if all vortex terms are irrelevant.

Without imposing any global symmetries, the mass terms $\lambda_{e,1}$ and $\lambda_{m,1}$ are allowed, and they are relevant when $K < 2$ and $K > \frac{1}{2}$, respectively. We see that these two regions cover all of $K > 0$. Hence for any $K$, at least one of the two perturbation becomes relevant and gaps out the edge. Therefore, without any global symmetries, a generic $\mathbb{Z}_2$ topological state always has a gapped edge. In the extended BFG model, the nontrivial symmetry fractionalization forbids certain vortex terms in Eq. (3) and therefore allows the edge to remain gapless in a finite region of $K$. To be specific, we show in the Appendix C that $R_\pi^2 = -1$ for spinons leads to $p \geq 2$ and $(MT)^2 = -1$ for both spinons and visons implies $p, q \geq 2$. Therefore, inside the region $1/2 < K < 2$, the edge is gapless. When $K > 2$, the visons will condense, breaking the composite symmetry $MT$: in terms of spin order, this turns out to be an Ising antiferromagnetic order with magnetization along $z$-axis ($z$-Ising). When $K < 1/2$, the spinons condense, breaking $R_\pi$: this is an Ising ferromagnetic order with magnetization along $x$-axis ($x$-Ising). A phase diagram for the symmetric edge is plotted in Fig.2(a). For completeness, we remark that when U(1)-symmetry is restored, it forbids all vortex terms for spinons, and the gapless region becomes larger $0 < K < 2$. [This is consistent with the fact that the U(1) symmetry cannot be spontaneously broken on the 1D edge.] Hence, in our study, we explicitly break the U(1) symmetry, to concentrate on gapless edge states solely protected by the $MT$ symmetry.

Can we still have a gapless edge when $MT$ is broken? The edge theory analysis gives a negative answer. When $MT$ broken while $R_\pi$ is preserved, the $\lambda_{e,2}$-term for spinons and the $\lambda_{m,1}$-term for visons are allowed, which respectively become relevant for $K > 1/2$ and $K < 1/2$, again ruling out any room for gapless edge excitations (see Appendix C for more details). The phase diagrams for the symmetry breaking scenarios are plotted in Fig. 2(b).

By analyzing the edge theory, we find that two entries in the fracitonalization data, namely, $(MT)^2 = -1$ for both the spinon and the vison, guarantee gaplessness on a symmetric edge; and we confirm that any gapped edge must break $MT$.

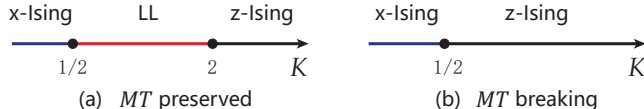

FIG. 2. Schematic edge phase diagram. In $x$-Ising phase, spinons condense on the edge; in $z$-Ising phase, visons condense on the edge. LL is the Luttinger liquid phase with gapless edge excitations, protected by $MT$ symmetries.

## IV. NUMERICAL RESULTS

We employ large-scale quantum Monte Carlo simulations to verify the spin liquid in the bulk, and more importantly, the existence of symmetry protected edge modes. As shown in Fig. 1 (a), to investigate the symmetry protected edge states, we choose open boundary condition in $y$-direction, i.e., a ribbon geometry of the kagome lattice, with the ribbon size $N = 6L_x L_y + L_y$, where $L_x$ is the length, $L_y$ is the width, and there are 6 sites per unit cell. We implement a finite-temperature Stochastic Series Expansion (SSE-QMC) alogrithm with directed loop update [33]. Since the model is highly anisotropic and frustrated, i.e., $J_z \gg J$ and $J_z \gg J'$, the energy landscape in the configuration space is complicated with many local minima. To overcome the hence induced sampling problem, we perform the QMC update with a 4-spin operator as a plaquette (8 legs in a vortex) [34], instead of the conventional 2-spin operator. Moreover, to reduce the rejection rate of the proposed spin configuration, we make use of a specific algorithm that satisfies the balance condition without imposing detail balance in the Markov chain of Monte Carlo configurations [35]. To our knowledge, this is the first time such advanced Monte Carlo algorithms (plaquette update and balance condition) have been combined together to investigate highly frustrated spin systems that host spin liquid ground state. Details of our efficient numeric method are presented in the Appendix A .

We set $J_z = 1$ as the energy unit, which is the largest energy scale of the problem, perform simulations with the ribbon size $L_x = 3L_y$, and $L_y = 8$, 10 and 14, and set the inverse temperature at $\beta = 500$ and find in the parameter space a half-dome (the blue volume in Fig. 1 (b)) under which there is a single phase without symmetry breaking (see Appendix B for the discussion of bulk phase boundary). Since this region includes part of the $J/J_z$-axis identified by previous QMC simulations as $\mathbb{Z}_2$ spin liquid state [29–31], and there is no signature of phase transition inside the half-dome from our simulation result, we conclude that the entire phase has the same identity.

We choose a representative parameter set $J = J' = 0.05$ in the $\mathbb{Z}_2$ spin liquid [labeled "×" in Fig. 1(b)], and study the edge states therein. In light of the previous analysis, we expect that gapless edge modes exist within some region of $K$. However, no explicit relation is known

between $K$ in the effective theory and the physical parameters that can be tuned in a simulation. A "clue" for tuning comes from the realization that the spinon-condensed edge (small $K$) is $x$-Ising ordered and the vison-condensed edge (large $K$) is $z$-Ising ordered. On the other hand, we notice that on the edge, the $J$ term and the $J_z$ term favor $x$-Ising order and $z$-Ising order respectively. Therefore, we expect that by tuning $J^e$ and $J'^e$ (the coupling constants on the edge), leaving bulk parameters intact, we can effectively tune $K$, thus scanning the phase diagram on the edge, as proposed in Fig. 2.

The results are summarized in Fig. 3. To check whether the edge is in a certain phase, we calculate the spin-spin correlation function $G(r) \equiv \langle S_0^+ S_r^- \rangle$ along the edge of the kagome ribbon, plotted in Fig. 3(a). We identify three edge phases separated by two critical $J_{c1}^e = 0.51$ and $J_{c2}^e = 0.56$: (i) for $J^e < J_{c1}^e$, $G(r)$ is short-ranged; (ii) for $J_{c1}^e < J^e < J_{c2}^e$, $G(r)$ decays algebraically; and (iii) for $J^e > J_{c2}^e$, $G(r)$ becomes long-ranged and converges to a finite value at long distance. Further QMC simulations show that in phase-(i), the longitudinal correlation $\langle S_0^z S_r^z \rangle$ decays to a finite value at large distance; in phase (ii) $\langle S_0^z S_r^z \rangle$ decays algebraically as well, and in phase-(iii), $\langle S_0^z S_r^z \rangle$ is short-ranged. Here we show the comparison of both transverse and longitudinal spin-spin correlation functions $G(r) = \langle S_0^+ S_r^- \rangle$ and $C(r) = \langle S_0^z S_r^z \rangle$. Fig. 4 (a) and (b) depicts the situation as in Fig. 3 (a). We have chosen $J^e = 0.48$ (inside the phase-(i), $J^e < J_{c1}^e$), $J^e = 0.53$ (inside the phase-(ii), $J_{c1}^e < J^e < J_{c2}^e$) and $J^e = 0.57$ (inside the phase-(iii), $J_{c2}^e < J^e$). It is clear that while $G(r)$ is short-ranged, $C(r)$ is long-ranged, hence the edge is in the $z$-Ising phase (phaes-(i)), and while $G(r)$ is power-law decay, $C(r)$ is power-law decay as well, hence the edge is in the gapless phase with $MT$-protected edge states (phase-(ii)), and while $G(r)$ is long-ranged, $C(r)$ is short-ranged, and the edge is in the $x$-Ising phase. Fig. 4 (c) further demonstrates that in the $z$-Ising phase, the $C(r)$ is antiferromagnetic in nature. It is thus reasonable to identify the three phases with $z$-Ising ordered (phase-(i)), gapless (phase-(ii)), and $x$-Ising ordered (phase-(iii)), accordingly, consistent with edge analysis in Fig. 2 (a). It is easy to see that in both Ising-ordered phases the symmetry $MT$ is broken.

To verify that the gapless edge modes are indeed protected by the symmetry $MT$, we explicitly break the symmetries on the edge in the following ways by putting additional terms on the edge. In Fig. 3(b), we break time-reversal while keeping mirror reflection symmetry, by coupling the spins on the edge to an extra Zeeman field along $z$-axis, $h^e S_{i \in \text{edge}}^z$, with $h^e = 1$. In Fig. 3(c) we break mirror reflection symmetry while keeping time-reversal, by modifying the coupling $J^e$ on the edge in a dimerizing pattern, i.e, $J^{AB} = J^{CD} = J^e + 0.1$ and $J^{BC} = J^{DA} = J^e - 0.1$. In both cases, we break the composite symmetry $MT$, and find that the region where $G(r)$ algebraically decays vanishes. This indicates that with $MT$ broken, the edge becomes fully gapped.

For further comparison, in Fig. 3(d), we put on the

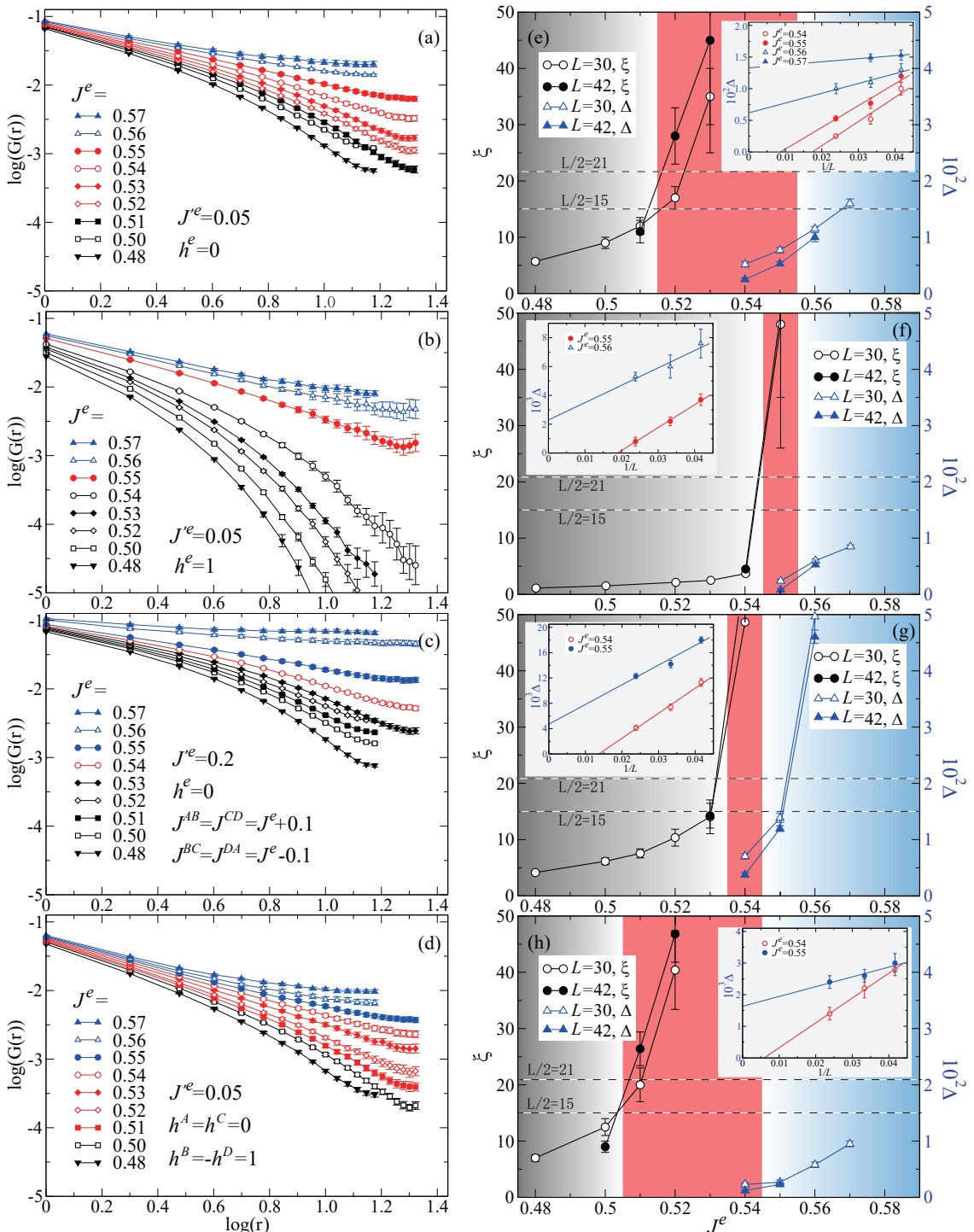

FIG. 3. Equal time transverse spin-spin correlation $G(r) = \langle S_0^+ S_r^- \rangle$ as a function of distance $r$ along the edge. (a) $MT$ is preserved. Here one sees as a function of $J^e$, $G(r)$ changes from exponential decay to long-range order, and in between, there is a finite critical region of $J^e$ from $0.51 < J^e < 0.56$, that $G(r)$ has power-law decay. This finite critical region is shown in the red area in (e). (b) $MT$-symmetry is broken by the uniform magnetic field $h^e = 1$. $G(r)$ has either exponential decay or long-range order as a function of $J^e$, the edge states vanish, and the critical point is close to $J^e = 0.55$, as also shown in (f). (c) $MT$-symmetry is broken by the different edge-spin coupling $J^{AB} = J^{CD} = J^e + 0.1$ and $J^{BC} = J^{DA} = J^e - 0.1$. $G(r)$ has either exponential decay or long-range order as a function of $J^e$, the edge states vanish, and the critical point is close to $J^e = 0.54$, as also shown in (g). (d) Both $M$- and $T$-symmetries are broken by staggered magnetic field $h^A = h^C = 0$ and $h^B = -h^D = 1$, but the product of $MT$ is preserved. As a function of $J^e$, in between exponential decay and long-ranged order, there is a finite critical region of $0.50 < J^e < 0.55$, that $G(r)$ has power-law behavior, i.e., there exists edge states. This finite critical region is shown in the red area in (h). (e)-(h) are the quantitative analyses of the $G(r)$ data. The $\xi$ is the correlation length obtained from Eq. 4 and $\Delta$ is the expectation value of $\langle S_0^+ S_r^- \rangle$ at long distance, i.e., the square of $x$-Ising order parameter, obtained from Eq. 5. The insets are finite size scaling of $\Delta$.

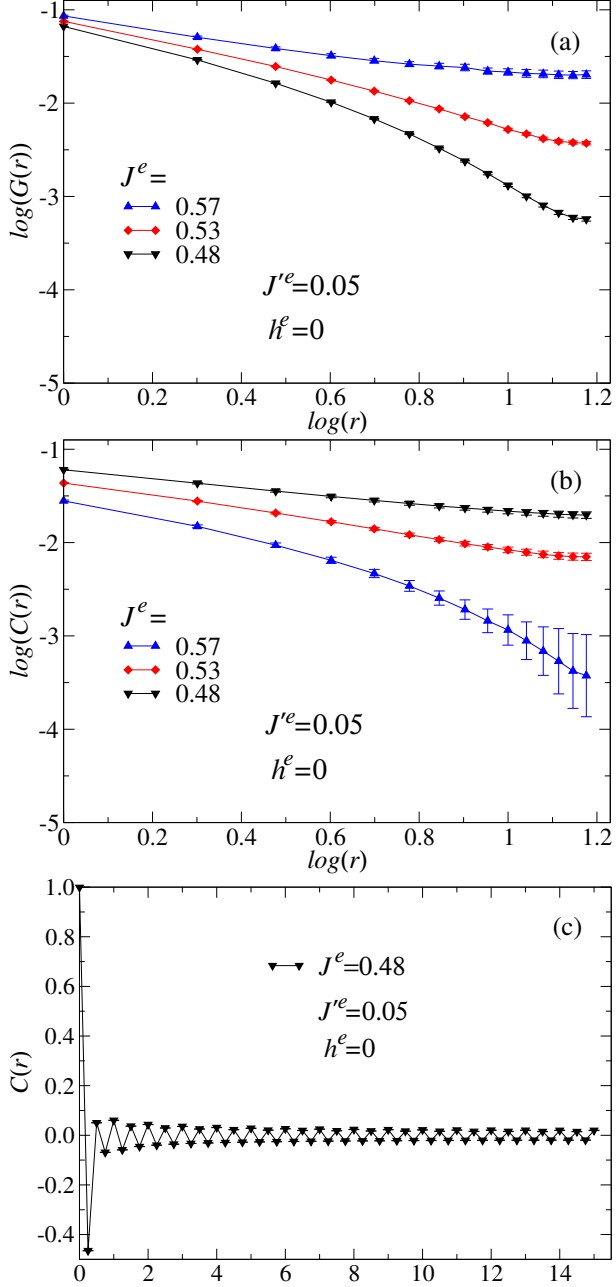

FIG. 4. (a) Transverse spin-spin correlation $G(r) = \langle S_0^+ S_r^- \rangle$ and (b) longitudinal spin-spin correlation $C(r) = \langle S_0^z S_r^z \rangle$ as funtion of distance $r$ on the edge. The parameters are chosen as those in Fig. 3 (a). (c) The $z$-Ising anti-ferromagnetic order in phase-(i).

edge a staggered Zeeman field, $h^A = h^C = 0$ and $h^B = -h^D = 1$, which breaks both $M$ and $T$ but preserves $MT$, and find that in this case, the gapless region persists within a window $J_{c1}^e < J^e < J_{c2}^e$ where $J_{c1}^e = 0.50$ and $J_{c2}^e = 0.55$. This confirms the predicted phase diagram in Fig. 2, and implies that the gapless edge is protected by the composite $MT$ but not individually by $M$ and $T$.

To determine the phase boundaries quantitatively in

Fig. 3 (a), (b), (c) and (d), we furthermore fit the decay of the $G(r)$ to distinguish the phases (i), (ii) and (iii). For example, in Fig. 3 (e), we fit the data with

$$G(r) = a \times r^{-b} \times \exp(-r/\xi), \qquad (4)$$

where $a$, $b$ and $\xi$ are fitting parameters, which has been successfully applied in similar situation [36]. As shown in Fig. 3 (e), when $J^e$ is tuned towards $J_{c1}^e$, the correlation length $\xi$ increases drastically. At $J_{c1}^e$, the correlation length becomes larger than half of the ribben length ($L/2$, $L$ is the $L_x$), signifying that the edge changes from exponential decay to power-law decay. The boundary $J_{c2}^e$ is harder to determine, as it is the separation of power-law decay and long-range $x$-Ising order. To this end, we fit the data with

$$G(r) = \Delta + a \times r^{-b} \times \exp(-r/\xi), \qquad (5)$$

where $\Delta$ would be the order parameter square if there were $x$-Ising order. In the main panel of Fig. 3 (e), one can see that close to $J_{c2}^e$, the finite size effect of $\Delta$ for $L = 30$ and 42 is strong. Hence, in the inset of Fig. 3 (e), a $1/L$ finite size scaling of $\Delta$ is applied. With $L = 24, 30$ and 42, one clearly sees that at $J^e = 0.55 < J_{c2}^e$ there is still no long-range order of $x$-Ising along the edge. And only when $J^e = 0.56 \sim J_{c2}^e$, the $x$-Ising order manifests. Hence, the quantitative phase boundaries determined from fitting the correlation function with Eq. 4 and 5, in Fig. 3 (e) agree well with those observed in Fig. 3 (a), with the red region in Fig. 3 (e) as the critical phase on the edge.

Similar analyses have been applied in Fig. 3 (f) and (g), to quantify the single phase transition between $z$-Ising and $x$-Ising phases. As shown in the Fig. 3 (f) and (g), as the $J^e$ increases from $z$-Ising phase towards $x$-Ising phase, the correlation length $\xi$ fitted from Eq. 4 increases, and it suddenly becomes larger than $L/2$ at the phase transition. $J^e = 0.55$ for Fig. 3 (f) and $J^e = 0.54$ for Fig. 3 (g). From this point on, we also fit the correlation function with Eq. 5, to determine the $x$-Ising order parameter square $\Delta$. The insets of Fig. 3 (f) and (g) show the finite size scaling ($1/L$ scaling) of the order parameter square. It is clear that the $x$-Ising order develops right above $J^e = 0.55$ for Fig. 3 (f) and $J^e = 0.54$ for Fig. 3 (g). These results clearly demonstrate that once the $TM$ symmetry is broken, there is the no more intermediate phase with power-law decay of edge states, as in Fig. 2 (a). The situation goes back to Fig. 2 (b).

Finally, the analysis in Fig. 3 (h) is for the phase boundaries in Fig. 3 (d). The correlation length $\xi$ goes beyond $L/2$ at $J_{c1}^e$, and in between $J_{c1}^e < J^e < J_{c2}^e$, the $x$-Ising long-range order is absent, as shown in the inset of the finite size scaling. Again, the red area in Fig. 3 (h) is fully consistent with the phase boundary shown in Fig. 3 (d).

## V. DISCUSSION

We have shown that the symmetry fractionalization pattern in the $\mathbb{Z}_2$ spin liquid ground state of the modified BFG model implies the existence of symmetry protected edge modes. This model is one of the few known models possessing the spin liquid ground state and consists of no more than two-body interactions preserving U(1) spin rotation symmetry, and hence may be related to realistic compounds such as herbertsmithite [37, 38], $\alpha$-$R$Cl$_3$ [39], where $R$ is some rare-earth element as well as the recently synthesized kagome quantum spin liquid compound Zn-doped barlowite Cu$_3$Zn(OH)$_6$FBr [40, 41]. (A caveat is due, however, that this model cannot be adiabatically connected to gapped symmetric spin liquids with full SU(2) spin rotation symmetry on kagome lattice [27].) From a phenomenological point of view, this topological spin liquid has two key features that (i) the bulk is symmetric and gapped (no gapless excitations) and (ii) the edge is gapless if and only if there is no symmetry breaking, and both features can be tested in experiments. In a transport measurement of heat or spin, it is expected that the static longitudinal conductance is independent of the width of the sample, as the carriers go along the edge but not through the bulk. One may also use magnetic force microscope to locally probe the spin gap on the edge or in the bulk, and it is expected that in the AC mode, the quality factor of the cantilever sees an additional drop on the edge due to damping from the edge modes. For completeness, we remark that other states with gapped bulk and gapless edge are the symmetry-protected topological phases [42], but it can be shown that with $M$ and $T$ symmetries, on the edge there is always an instability towards a symmetry breaking phase (see Appendix D and Ref. 43 for details), and so is essentially different from the spin liquid case discussed in this paper.

In summary, we show that gapped topological spin liquids may have symmetry fractionalization patterns that lead to symmetry-protected edge modes. These edge modes can be utilized as characteristic feature of the underlying spin liquids and give us new clues for their experimental identification.

### ACKNOWLEDGEMENT

We thank Jinxing Zhang for comments on the magnetic force microscope, Xiao Yan Xu for comments on the manuscript, and A. W. Sandvik as well as H. Suwa for discussion on the plaquette update and balance condition. We (Y.C.W., C.F. and Z.Y.M.) acknowledge fundings from the Ministry of Science and Technology of China through National Key Research and Development Program under Grant Nos. 2016YFA0300502 and 2016YFA0302400 and the National Science Foundation of China under Grant Nos. 11421092, 11574359 and 11674370, as well as the National Thousand-Young-Talents Program of China. We thank the Center for Quantum Simulation Sciences in the Institute of Physics, Chinese Academy of Sciences; the Tianhe-1A platform at the National Supercomputer Center in Tianjin for their technical support and generous allocation of CPU time.

## Appendix A: Generalized SSE-QMC algorithm

Here we describe in detail the SSE-QMC simulation with plaquette update as well as the balance condition. This part follows closely to the Refs. 34 and 35.

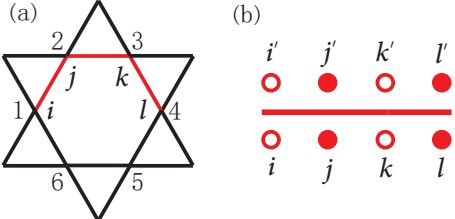

FIG. 5. (a) A plaquette with 4 sites $i, j, k, l$ on the kagome lattice. (b) A vortex with 8 legs.

The first step is to rewrite the Hamiltonian as a sum of elementary interactions,

$$H = -\sum_a \sum_b H_{a,b} \quad (A1)$$

where the index $a$ refers the operator types: $a = 1$ is the diagonal operator in the $S^z$ basis and $a = 2$ is the off-diagonal operator in the $S^z$ basis, and $b$ is the lattice unit over which the interactions are summed. Due to the highly frustrated and anisotropic form of Hamiltonian in Eq. 1, i.e., $J \ll J_z$ and $J' \ll J_z$, we take 4-site plaquette as a lattice unit in the QMC update (8 legs in a vortex in the SSE language) and consider all the interactions within this plaquette as one entity, as show in Fig. 5. This treatement is different from the conventional 2-site bond operator updates (4 legs in a vortex)[33] in SSE, and it is shown both in Ref. 34 and in Fig. 6 (will discuss below) that for the highly frustrated and anisotropic spin systems like ours, plaquette update is the only way to ensure efficiency in the update and ergodicity in the Markov-chain of configurations.

By taking this decomposition, the diagonal term $H_{1,b}$ and off-diagonal term $H_{2,b}$ can be written as

$$H_{1,b} = C - \frac{J_z}{z_1} \left( S_i^z S_j^z + S_j^z S_k^z + S_k^z S_l^z \right)$$
$$- \frac{J_z}{z_2} \left( S_i^z S_k^z + S_j^z S_l^z \right) - \frac{J_z}{z_3} S_i^z S_l^z \quad (A2)$$
$$+ \frac{h}{z} \left( S_i^z + S_j^z + S_k^z + S_l^z \right),$$

$$H_{2,b} = \frac{J}{z_1} \left( S_i^+ S_j^- + S_j^+ S_k^- + S_k^+ S_l^- + h.c. \right), \quad (A3)$$

where the site indices $i, j, k, l$ within a plaquette are illustrated in Fig. 5 (a), and $C$ is some constant necessary to keep the corresponding transition probabilities positive definite (and hence avoid the sign problem). In order to cover full interactions of the lattice by the 4-site plaquette, there should be 6 plaquettes for each hexagon, i. e., $\{i, j, k, l\} = \{1, 2, 3, 4\}, \{2, 3, 4, 5\}, ..., \{6, 1, 2, 3\}$. In periodical boundary condition, $z_1 = 3, z_2 = 2, z_3 = 2$ are the times of recounting interaction and $z = 8$ are the times of recounting sites.

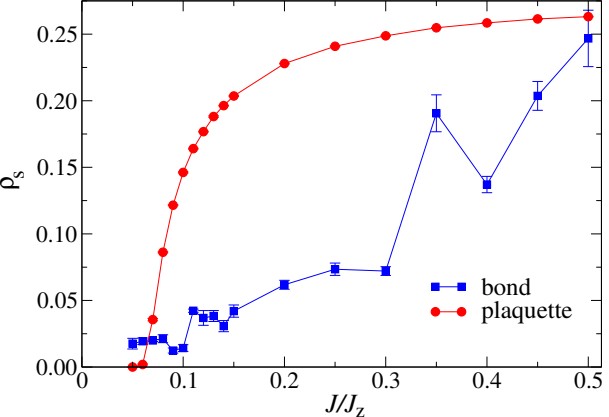

FIG. 6. Spin stiffness $\rho_s$ as function of $J/J_z$ for our model in Eq. 1 with the $N = 6 \times 6 \times 3$ at $h = 0$, $J'/J_z = 0$ and $\beta J/J_z = 12$. $10^6$ QMC measurements were performed. "plaquette" stands for the plaquette update and balance condition implemented in our SSE-QMC simulations and "bond" stands for the conventional bonds version of SSE algorithm.

The next step is to implement solutions of the directed loop equations. We implement the Markov chain Monte Carlo method with balance condition (without detail balance condition) developed by Suwa and Todo [35], to easily obtain the solutions of the direct loop equations. A vortex $\alpha$ with 8 legs (as shown in Fig. 5 (b)), whose configurational weight is $W_\alpha$, can be updated to vortex $\beta$, whose configurational weight is $W_\beta$, with a transition probability $p_{\alpha \to \beta}$ within a loop update. By introducing a quantity $V_{\alpha \to \beta} := W_\beta p_{\alpha \to \beta}$ for convenient, the detail balance condition is to require $V_{\alpha \to \beta} = V_{\beta \to \alpha}$. But this turns out to be too rigid a requirement to construct the directed loop equations. In fact, as proved in Ref. 35, the balance condition $W_\beta = \sum_\alpha V_{\alpha \to \beta}$ (for any $\beta$) combined with the law of probability conservation $W_\alpha = \sum_\alpha V_{\alpha \to \beta}$ (for any $\alpha$) are enough to implement Markov chain Monte Carlo algorithm and provide solutions to the directed loop equations. Suppose $W_1 = \max\{W_\alpha\}$, the solutions with minimum average rejection rate can be given as,

$$V_{\alpha \to \beta} = \max(0, \min(\Delta_{\alpha\beta}, W_\alpha + W_\beta - \Delta_{\alpha\beta}, W_\alpha, W_\beta)), \quad \text{(A4)}$$

where

$$\Delta_{\alpha \to \beta} = S_\alpha - S_{\beta-1} + W_1, 1 \le \alpha, \beta \le 8, \quad \text{(A5)}$$

$$S_\alpha := \sum_{k=1}^{\alpha} W_k, 1 \le \alpha \le 8, \quad \text{(A6)}$$

$$S_0 := S_8. \quad \text{(A7)}$$

We use Eq. A4 to easily obtain the solutions of the directed loop equations and to hence constructed directed loop update with minimal average rejection rate.

To demonstrate the power and necessity of plaquette update and balance condition, in Fig. 6, we compare the results of spin stiffness $\rho_s$ (superfluid density in hard-core boson description) as function of $J/J_z$ for our model in Eq. 1 on the $L = 6$ system. It is clear that only the plaquette update and balance condition provide the correct results of $\rho_s$ as $J/J_z$ is going towards the FM to $\mathbb{Z}_2$-SL critical point at $J/J_z = 0.071(1)$. The conventional bond version of the SSE algorithm provide wrong results precisely because the simulations suffer efficiency as well as ergodicity problem for the highly frustrated and anisotropic system like ours.

## Appendix B: Bulk phase diagram

To determine the three dimensional phase diagram as shown in Fig. 1 (b), we calculate the expectation value of $E'_k$ (it is the kinetic energy in the hard-core boson description of our Hamiltonian),

$$E_{k'} = -\frac{\langle N_2 \rangle}{\beta N}, \quad \text{(B1)}$$

where $N_2$ is the number of $S_i^\pm S_j^\pm$ operators in a configuration and $N = 3L^2$ is the system volume. $E_{k'}$ is the first order derivative of the free energy over the control parameter of the phase transition: $\frac{\partial \langle H \rangle}{\partial J'}$ when $J$ is fixed. So with $E_{k'}$ one can tell both the phase boundary and the order of the phase transition while changing $J'$.

We calculate $E_{k'}$ at $\beta(J + J') = 2L$ with system size $L = 18$ in periodical boundary condition. The main results are shown as Fig. 7 (a). Clearly, there is a first order phase transition as a function of $J'/J_z$ for fixed $J/J_z$ (although the order of the transition is not so clear at $J/J_z = 0.06$, but the transition point can still be determined). Taking these results, we can draw the phase boundary of our extended BFG model at $h = 0$ shown as Fig. 7(b).

## Appendix C: Edge theory

In this section, we derive the edge theory in the section III in full clarity.

We consider how $\theta$ and $\phi$ transform under $Z$, $M$ and $T$, in a way that is consistent with the symmetry fractionalization class. It is easy to check that the transformations

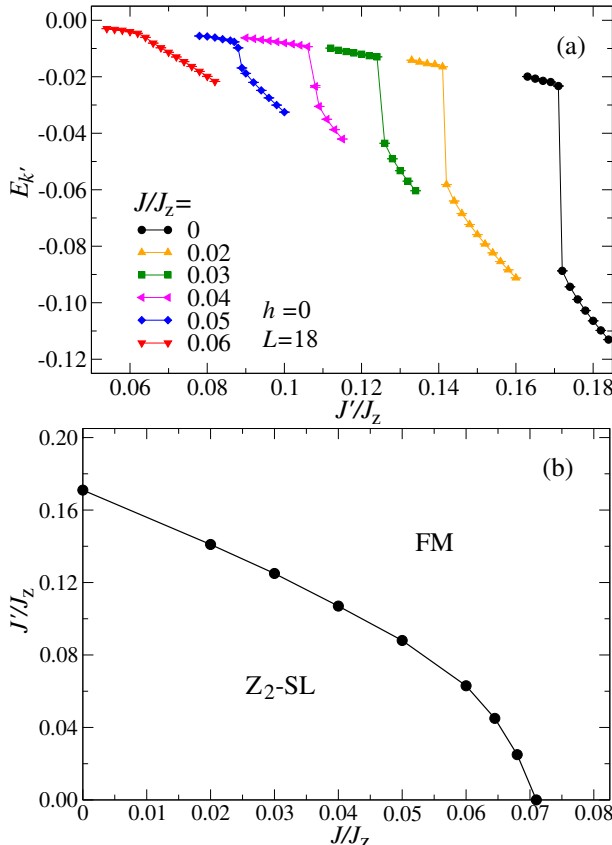

FIG. 7. (a) $E_{k'}$ as function of $J'/J_z$ for each fixed $J/J_z$. (b) Phase boundary of the extended BFG model at $h/J_z = 0$.

listed in Table II satisfy the symmetry fractionalization in the main text and keep the Luttinger liquid Lagrangian Eq. 3 invariant.

Using these results, we can derive how the vortex terms $\cos(2\phi + \alpha_e)$ and $\cos(2\theta + \alpha_m)$ transform under symmetries, and the results are also listed in Table II. As expected, $\cos(2\phi + \alpha_e)$ changes sign under $Z$, $T$ and $MT$, so $\lambda_{e,1}$ is forbidden by either $Z$, $T$ or $MT$ (in the case of $MT$, if $\lambda_{e,1}$ is an odd function of $x$ the term is again symmetric, but then $\lambda_e(x = 0) = 0$, still leaving some unbroken degeneracy on the edge). Similarly, $\cos(2\theta + \alpha_m)$ changes sign under $MT$, and $\lambda_{m,1}$ is forbidden by $MT$ (we notice that $\lambda_{m,1}$ is not forbidden by either $T$ or $M$ alone).

In summary, in the extended BFG model we considered here, both $\lambda_{e,1}$ and $\lambda_{m,1}$ are forbidden by the $MT$ symmetries. When this happens, $\lambda_{e,2}$ and $\lambda_{m,2}$ are the most relevant terms, and they are relevant when $K < \frac{1}{2}$ and $K > 2$, respectively. Hence, there is a finite range of $\frac{1}{2} < K < 2$ where the gapless Luttinger liquid phase is stable. The stability of this gapless edge state is protected by the nontrivial symmetry fractionalization pattern.

Now we discuss the phase diagram of the edge states when symmetries are broken. We will focus on the $T$ and $M$ symmetries. To understand the symmetry-breaking

phases, we first need to know how physical observables are represented in the edge theory. This is already hinted above, but now we make it more explicit. Since $e^{i\phi}$ creates $e$ anyons which carry spins, the spin density is given by the dual variable:

$$S^z = \frac{1}{2\pi} \partial_x \theta. \tag{C1}$$

One can then naturally associate $e^{\pm 2i\phi}$ to $S^\pm$. Therefore, if $e^{i\phi}$ condenses the edge has an Ising magnetic order with magnetization in the XY plane. On the other hand, if $e^{i\theta}$ orders depending on details the edge can form a $S^z$ order (e.g. antiferromagnetic) or a valence-bond solid. The phase of the edge state is determined by the value of the Luttinger parameter $K$, which can be tuned for example by changing the hopping amplitude on the edge. For instance, $K$ decreases as one increases the hopping amplitude, or decreases the repulsive interaction.

1. Breaking $T$ but not $M$. Keeping the most relevant perturbations:

$$\mathcal{L}_v = \lambda_{e,2} \cos(4\phi + \alpha) + \lambda_{m,1} \sin 2\theta. \tag{C2}$$

When $K < 1/2$, the $\cos(4\phi + \alpha)$ becomes relevant and drives a transition to XY order (happens when hopping amplitudes on the edge are increased, which decreases $K$). When $K > 1/2$, $\sin 2\theta$ becomes relevant and drives a transition to a $S^z$ order. The $K = 1/2$ is the Kosterlitz-Thouless transition.

2. Breaking $M$ but not $T$. The most relevant perturbations are

$$\mathcal{L}_v = \lambda_{e,2} \cos(4\phi + \alpha) + \lambda_{m,1} \cos 2\theta. \tag{C3}$$

The results are similar to the previous case, with two gapped phases separated by a transition at $K = 1/2$.

## Appendix D: Edge states of symmetry-protected topological phase

Symmetry-protected phase also exhibits a gapped bulk with gapless edge states protected by symmetry. We will follow the approach of Ref. 44 and describe the edge of a SPT phase in terms of a Luttinger liquid theory:

$$\mathcal{L} = \frac{i}{4\pi} \partial_\tau \phi \partial_x \theta - \frac{u}{2\pi} \left[ K(\partial_x \theta)^2 + \frac{1}{K}(\partial_x \phi)^2 \right]. \tag{D1}$$

We choose a reflection-invariant edge, and under $M$ we have $x \to -x$. It is known that either $M$ or $T$ alone does not protect nontrivial SPT phases, but with both $M$ and $T$ there are three nontrivial phases. They are generated by two root phases:

TABLE II. Symmetry transformation of operators in the Luttinger liquid edge theory.

| Field | $Z$ | $T$ | $M$ | $MT$ |
|---|---|---|---|---|
| $\phi(x)$ | $\phi(x) + \frac{\pi}{2}$ | $\phi(x) + \frac{\pi}{2}$ | $\phi(-x)$ | $\phi(-x) + \frac{\pi}{2}$ |
| $\theta(x)$ | $\theta(x)$ | $-\theta(x)$ | $-\theta(-x) + \frac{\pi}{2}$ | $\theta(-x) + \frac{\pi}{2}$ |
| $\cos[2\phi(x) + \alpha_e]$ | $-\cos[2\phi(x) + \alpha_e]$ | $-\cos[2\phi(x) + \alpha_e]$ | $\cos[2\phi(-x) + \alpha_e]$ | $-\cos[2\phi(-x) + \alpha_e]$ |
| $\cos[2\theta(x) + \alpha_m]$ | $\cos[2\theta(x) + \alpha_m]$ | $\cos[2\theta(x) - \alpha_m]$ | $-\cos[2\theta(-x) - \alpha_m]$ | $-\cos[2\theta(-x) + \alpha_m]$ |

#### 1. Phase I.

$$M : \begin{pmatrix} \phi(x) \\ \theta(x) \end{pmatrix} \rightarrow \begin{pmatrix} \phi(-x) + \pi \\ -\theta(-x) \end{pmatrix}$$
$$T : \begin{pmatrix} \phi(x) \\ \theta(x) \end{pmatrix} \rightarrow \begin{pmatrix} -\phi(x) \\ \theta(x) + \pi \end{pmatrix} \tag{D2}$$

In fact, in this case the edge is protected by $MT$ alone.

#### 2. Phase II.

$$M : \begin{pmatrix} \phi(x) \\ \theta(x) \end{pmatrix} \rightarrow \begin{pmatrix} -\phi(-x) + \pi \\ \theta(-x) \end{pmatrix}$$
$$T : \begin{pmatrix} \phi(x) \\ \theta(x) \end{pmatrix} \rightarrow \begin{pmatrix} -\phi(x) \\ \theta(x) + \pi \end{pmatrix} \tag{D3}$$

In both cases, the minimal symmetry-allowed vortex terms are

$$\mathcal{L}_v = \lambda_\phi \cos 2\phi + \lambda_\theta \cos 2\theta. \tag{D4}$$

The scaling dimensions are $2K$ and $\frac{2}{K}$, respectively. So except the marginal point $K = 1$, everywhere else either of the two becomes relevant and the edge spontaneously breaks the symmetries. In fact, in a sense the instability of the Luttinger liquid edge is the common feature of bosonic SPT phases protected by order-2 symmetries (i.e. the Levin-Gu SPT phase protected by an on-site $\mathbb{Z}_2$ symmetry). So adding the Ising symmetry $R_\pi$ does not change the conclusion.

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
