# Peer review of "Topological Spin Liquid with Symmetry-Protected Edge States"

_SciPost Physics_

## Round 2 · Referee Report · Anonymous (Referee 1) · 2018-5-11

Strengths

  1. Interesting conclusion that the edge must be gapless if time-reversal and mirror reflection symmetries are both preserved.
  2. Combined theoretical and numerical study.

Weaknesses

  1. The model seems to exhibit gapless edge excitations only in narrow parameter regime of boundary couplings, different from the bulk couplings, which raises the question how relevant the findings are for real materials. This point / limitation does not seem to be discussed in the paper.
  2. The support for an extended gapless region from the numerics is not fully convincing (the left phase boundary in Fig.3(e), see report).

Report

In this paper the authors study a Z_2 toric-code like spin liquid on the Kagome lattice, where they find gapless edge modes which carry fractional quantum numbers (in a narrow parameter range of boundary couplings which are different from the bulk couplings). From a Luttinger liquid theory-based analysis on the edge they conclude that the edge must be gapless if time-reversal and mirror reflection symmetries are preserved. They support their findings by Quantum Monte Carlo simulations. One of the main motivations of this paper is to use the existence of protected, gapless edge modes as a probe for experiments to identify topological ordered phases.

This paper surely contains interesting material, e.g. the theoretical analysis that there is no gapless edge when the MT symmetry is broken which is also nicely supported by the QMC data. However, the parameter range where gapless edge modes are potentially stabilized seems very narrow, with properly chosen couplings on the boundary which are different from the couplings in the bulk. If the gapless edge modes only exist for fine-tuned boundary couplings, then the question is how relevant their findings are for real materials (where one cannot fine-tune the boundary couplings). At least from this model it seems rather unlikely that in a realistic setting one would observe gapless edge modes in a non-chiral Z_2 spin liquid, and therefore the experimental motivation to use the presence of gapless edge modes to identify topological ordered spin liquids does not seem very relevant. The fact that the boundary needs to be fine tuned in order to get the gapless edge modes is not mentioned in the abstract / conclusion which I find a bit misleading. The authors should discuss this issue and the limitations of their study in their paper, especially also in the discussion part where they try to connect their model to realistic compounds.

Concerning the numerics: it is generally very challenging to distinguish a power-law decay from an exponential one with a very long correlation length close to a critical point. (Also close to a critical point, the correlation length can increase with increasing system size since it gets arbitrarily large when approaching the critical point.) While 0.51 in Fig. 3(e) is definitely not critical, I do not see why 0.52 should be critical, rather than having a large correlation length (much larger than the system size). I also do not see how the 0.53 point can have a longer correlation length than half of the system. Could it be that to extract the correlation length all data points in G(r) have been included (also the ones where the correlation function starts flattening out, which would lead to an "overestimated" correlation length)?

It would be good if the authors could provide further support for the existence of an extended gapless region. For example, the authors could look at the vanishing of the z-Ising order parameter to determine the left phase boundary (similarly as they have determined the right boundary using the extrapolated x-Ising order parameter \Delta) and see if they obtain consistent results for the phase boundary.

To summarize, in my opinion the paper requires considerable revision of above-mentioned points before it is suitable for publication.

Requested changes

  1. Discussion of the relevance of their findings for real materials and experiments in view that the boundary couplings need to be tuned to a specific parameter range in order to get gapless edge modes.
  2. Further numerical support for the extended gapless region (shown in Fig.3(e), see report).

---

## Editorial Decision

awaiting_resubmission